# Soil Organic Nitrogen Components and N−Cycling Enzyme Activities Following Vegetation Restoration of Cropland in Danxia Degraded Region



Chao Wang [1,2,3], Qiannan Yang [1,2], Chi Zhang [3], Bo Zhou [4], Xiangdong Li [5], Xiaolong Zhang [1,2], Jing Chen [1,2] and Kexue Liu [1,2,*]

1　School of Resources and Planning, Guangzhou Xinhua University, Guangzhou 510310, China
2　Institute of South China Urban-Rural Economic and Social Development, Guangzhou 510642, China
3　College of Natural Resources and Environment, South China Agricultural University, Guangzhou 510642, China
4　Tea Research Institute, Guangdong Academy of Agricultural Science, Guangzhou 510640, China
5　Institute of Eco-Environmental and Soil Sciences, Guangdong Academy of Sciences, Guangzhou 510520, China
*　Correspondence: liukexue@xhsysu.edu.cn

**Abstract:** Soil organic nitrogen (SON) components are a key indicator of soil fertility and plant growth. The Danxia degraded region (DDR) is an ecologically fragile area in southern China, where the "Grain for Green" project has been implemented to prevent further land degradation. However, little is known about the effects of vegetation restoration on SON components in the DDR or the factors that influence them. We compared three vegetation restoration types, namely, grassland, shrubland, and arbor forest, with cropland to determine the relationship between SON components and N−cycling enzyme activities. Vegetation restoration increased the soil amino sugar N and amino acid N and reduced the proportion of non−hydrolyzable N. Compared with forest restoration measures, restoration to grassland was more beneficial to SON levels and N−fixation capacity. Vegetation restoration also increased soil nitrate reductase, denitrifying enzyme, protease, β−1,4−N−acetylglucosaminidase activities, and soil microbial biomass. Vegetation restoration in the DDR changed the SON components through the "mineralization−fixation" of organic matter via amorphous iron and proteases, which, in turn, affected the level of available soil N. Vegetation restoration improved the soil N structure and promoted the intrinsic soil N cycle, providing a scientific basis for soil quality restoration in the DDR.

**Keywords:** vegetation restoration; organic nitrogen components; N−cycling enzymes; Danxia; soil degradation

## 1. Introduction

Nitrogen (N) is a major limiting element in soils that affects plant growth and maintains ecosystem stability. It plays a key role in soil fertility and land productivity [1,2]. In the soil environment, over 90% of total nitrogen (STN) exists as organic N (SON), and the SON component and availability play an important role in soil N mineralization and supply [3,4]. Most SON must be degraded by microbial mineralization to produce mineral N and small molecule SON to meet the growth and reproduction needs of organisms [5,6]. Because of the complexity of SON components, active SON is both a substrate and a product of N mineralization [1]. Therefore, the active SON content can be an indicator of the potential of the soil N supply. However, excessive agricultural expansion had caused widespread destruction of vegetation, soil erosion, and land degradation globally, seriously threatening the ecological services of forest ecosystems and accelerating the loss of N [5].

Vegetation restoration is widely recognized as an effective strategy for regenerating degraded ecosystems by changing the composition and cover of vegetation communities

and increasing the net ecosystem productivity [6]. Vegetation restoration can also ultimately alter soil nutrient sequestration by modifying the microclimate and soil physical structure of the soil [7]. Vegetation converts inorganic N to SON, reduces soil N leaching losses, increases soil N inputs through root secretions and plant litter, and, in turn, promotes microbial growth and enhanced microbial N fixation [2,8]. Thus, vegetation restoration plays a key role in reducing soil erosion and promoting soil N sequestration. Although the effects of vegetation restoration on the SON/N content and storage of degraded soils were reported [8–10], data on the response of different active SON components are still limited. Soil chemical fractionation divides the SON pool into active SON (i.e., AAN, ASN, and AN) and recalcitrant SON (i.e., HUN and NHN), with different stability levels [11]. Active SON exhibits a rapid turnover rate, which includes fresh plant residuals, soil microbial exudates, and dead microorganisms [12,13]. They are regarded as an active N pool for rapid uptake and utilization by plants and soil microorganisms, which contributes to the turnover and circulation of soil nutrients [4,12,13]. Recalcitrant SON remains in the soil for long periods with a slow turnover rate, which is the key influencing factor for the long-term sequestration of SON [4,12]. Therefore, a comprehensive elucidation of the changes in active and recalcitrant SON components and their driving mechanisms during vegetation restoration have important implications for assessing SON pools and sequestration under the impact of vegetation restoration.

Nitrogen−cycling enzymes have specific biochemical catalytic capabilities and play a key role in maintaining the N homeostasis of ecosystems. Nitrate reductase (NRA) and denitrifying enzyme (DEA) facilitate the conversion between soil mineral N components [14]. These mineral N components can then be assimilated by microorganisms to form SON components for storage [15,16]. Protease (PT) activity is the rate-limiting step in SON mineralization, and thus, can be used as an indicator of N mineralization [17,18]. $\beta-1,4-N-$acetylglucosaminidase (NAG) plays an important role in the degradation of chitin and other glucosamine polymers [18]. Vegetation restoration was reported to increase N-cycling enzyme activities and reverse soil microbial N limitation in subtropical forest soil [9,19], which revealed that the effect of vegetation restoration could promote soil N turnover rates and increase the ability to supply N. Yan et al. [20] reported that N−cycling enzymes in grassland ecosystem soils increased significantly in the early stages of restoration, and remained stable as the restoration process progressed. These different results were probably due to differences in vegetation cover types, resulting in variations in plant litter, root systems and secretions, and soil properties [21–23]. Consequently, the effects of vegetation restoration on soil−N−cycling enzyme activities in a specific environment remain unclear.

The Danxia landform is a karst−like, red−bed landform that developed in the Tertiary red rock system and is mainly found in the southeastern subtropical humid zone of China (http://whc.unesco.org/en/list/1335/) (accessed on 9 October 2022) [24]. The Danxia landform has experienced long-term erosion by running water and the leaching of salt-based ions and nutrient elements from the soil. As a result, it formed a triadic structure of soil–rock–sparse vegetation, ultimately leading to the development of a unique ecologically fragile area in South China. Currently, land degradation in the Danxia landscape is a serious ecological problem in China [25,26]. To solve the problem of soil erosion and soil ecosystem degradation, a series of ecological restoration measures (returning cropland to grasslands and forests) have been implemented since the 1990s. Previous studies reported the response of N and soil enzyme activities to vegetation restoration on the Karst landform and the Loess Plateau [8–10]. However, little research has been conducted on the effects of vegetation restoration on SON components and N−cycling enzyme activities in the Danxia degraded region (DDR). Therefore, this study explored the effects of vegetation restoration on SON components in the DDR with two specific objectives: (1) to explore the differences in contents and stocks of SON components and soil−N−cycling enzyme activities with vegetation restoration and (2) to clarify the relationship between SON components and soil−N−cycling enzymes and the factors influencing the changes in the SON component.

## 2. Materials and Methods

### 2.1. Overview of the Study Area

The study area was in Danxianshan Global Geopark in Renhua County, Shaoguan City, Guangdong Province, China (Figure 1). This area has a subtropical monsoon climate with an average annual temperature of 19.9 °C. The average altitude of the study area is 220 m, with a Danxia landform composed of red glutenite layers and primarily Alfisols (red soil) [27]. Based on the degree of anthropogenic disturbance and the state of vegetation restoration, the soils of neighboring and environmentally similar cropland (control) and three types of restored vegetation (grassland, shrubland, and arbor forest), which are in different types of vegetation restoration, were selected for the study. The cropland (CL) has been cultivated for more than 30 years, mainly with *Brassica campestris* L. ssp. *chinensis* var. *utilis* Tsen, *Capsicum annuum* L., and *Zea mays* L., with strong anthropogenic disturbance and mainly inorganic chemical fertilizers. Grassland (GL) was formed by the fallowing of agricultural land, which is not managed and is weakly disturbed by human activities, with the main vegetation being Pas*palum thunbergii Kunth ex steud.* and *Setaria viridis* (L.). The shrubland (SL) and arbor forest (AF) have been restored for more than 30 years and are largely undisturbed, with the main vegetation being *Choerospondias axillaris*, *Schefflera octophylla*, and *Osteomeles subrotunda* in the SL, and *Firmiana danxiaensis*, *Castanopsis carlesii*, and *Castanopsis Carlesii* in the AF.

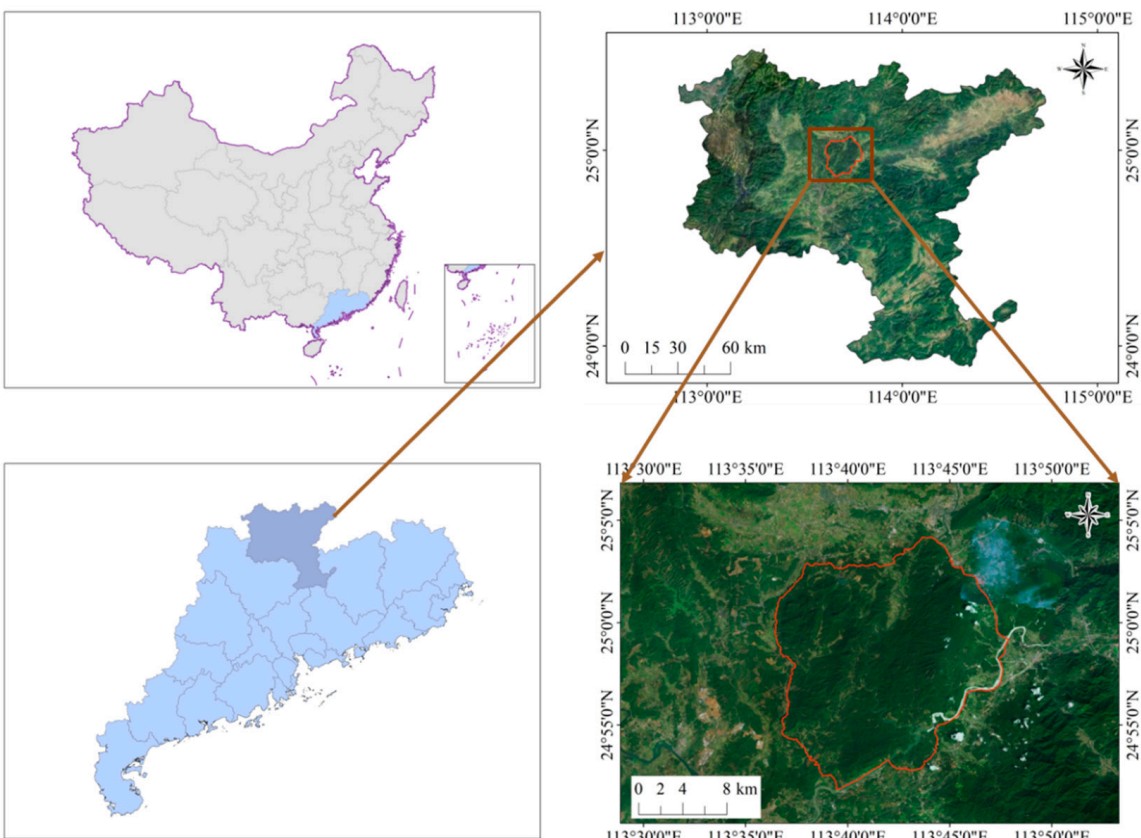

**Figure 1.** Location of Danxia Mountain, China.

### 2.2. Soil Sampling

Soil sampling plots of 10 m × 10 m were established in various vegetation areas, with three replicate plots for each vegetation type. Five replicate samples were collected along an "X" shaped area in each plot. The soil was sampled from the depth layers of 0–10 and 10–20 cm using an auger. The five replicate soil samples from the same soil depth were combined into one sample for each plot. Gravel and plant debris were removed from the soil samples, which were then divided into two portions. One portion was stored in a

sampling bag and transported to a laboratory, where it was naturally dried and ground through 10−and 100−mesh sieves for the determination of soil physicochemical properties. The other portion was collected in a self-sealing bag and stored in a refrigerator at 4 °C for the subsequent determination of microbial biomass carbon, nitrogen, and N−cycling enzyme activity. Some of the physicochemical properties of the soil samples are shown in Table 1.

**Table 1.** Selected soil physicochemical properties.

| Soil Depth (cm) | Vegetation Restoration | pH | BD (g cm$^{-3}$) | DOC (mg kg$^{-1}$) | MBC (mg kg$^{-1}$) | SOC (g kg$^{-1}$) | C:N | Fe$_o$ (g kg$^{-1}$) | Sand (%) | Silt (%) | Clay (%) |
|---|---|---|---|---|---|---|---|---|---|---|---|
| 0–10 | AF | 5.21 ± 0.04 c | 1.18 ± 0.01 c | 291.10 ± 5.95 a | 179.16 ± 8.97 a | 14.13 ± 0.81 b | 17.36 ± 1.22 a | 2.42 ± 0.13 b | 20.21 ± 0.94 c | 46.71 ± 2.08 b | 33.08 ± 2.34 a |
| | SL | 5.16 ± 0.17 c | 1.19 ± 0.00 b | 274.09 ± 12.81 ab | 157.10 ± 10.58 b | 24.32 ± 0.84 a | 7.32 ± 0.26 b | 1.72 ± 0.07 c | 23.16 ± 1.58 c | 45.06 ± 1.23 b | 31.79 ± 2.19 a |
| | GL | 6.28 ± 0.13 a | 1.15 ± 0.03 a | 275.82 ± 7.65 ab | 143.36 ± 10.44 b | 4.56 ± 0.39 c | 16.35 ± 0.95 a | 1.05 ± 0.06 d | 37.33 ± 4.47 a | 54.57 ± 4.85 a | 8.09 ± 1.13 c |
| | CL | 6.01 ± 0.12 b | 1.05 ± 0.02 a | 260.20 ± 14.75 b | 40.00 ± 3.04 c | 13.55 ± 0.56 b | 3.13 ± 0.34 c | 2.84 ± 0.07 a | 32.29 ± 1.30 b | 49.15 ± 1.19 b | 18.56 ± 0.14 b |
| 10–20 | AF | 4.80 ± 0.03 d | 1.20 ± 0.01 b | 230.69 ± 6.15 bc | 96.19 ± 8.09 a | 7.41 ± 0.56 b | 13.16 ± 1.44 a | 0.99 ± 0.07 d | 17.91 ± 1.22 c | 43.89 ± 1.51 c | 38.20 ± 0.29 b |
| | SL | 5.29 ± 0.02 c | 1.12 ± 0.01 b | 254.25 ± 15.27 a | 60.92 ± 2.37 b | 14.29 ± 0.84 a | 8.47 ± 0.33 b | 1.30 ± 0.09 b | 17.53 ± 0.19 c | 41.63 ± 0.52 c | 40.84 ± 0.35 a |
| | GL | 6.77 ± 0.15 a | 1.21 ± 0.01 a | 215.40 ± 8.25 c | 37.13 ± 5.65 c | 4.83 ± 0.44 c | 12.90 ± 1.39 a | 1.12 ± 0.02 c | 41.00 ± 1.76 a | 50.57 ± 1.77 b | 8.43 ± 0.42 c |
| | CL | 5.91 ± 0.06 b | 1.13 ± 0.02 a | 238.22 ± 11.04 ab | 28.31 ± 4.82 c | 8.41 ± 0.44 b | 3.94 ± 0.17 c | 2.81 ± 0.06 a | 30.00 ± 1.22 b | 66.19 ± 0.58 a | 3.81 ± 0.65 d |

BD, bulk density; DOC, dissolved organic carbon; MBC, microbial biomass nitrogen; SOC, soil organic carbon; C:N, ratio of SOC to STN; Fe$_o$, amorphous Fe. AF, arbor forest; SL, shrubland; GL, grassland; CL, cropland. Different lowercase letters indicate significant differences ($p < 0.05$) between vegetation restoration types at the same soil depth.

### 2.3. Soil Analysis

Soil physicochemical properties were determined according to Lu [28]. Soil pH was determined using the potentiometric method (soil–water ratio 1:2.5, *w*/*v*). Soil bulk density was determined by using the cutting ring method. Organic carbon was determined using the concentrated sulphuric acid–potassium dichromate external heating method. Total nitrogen was determined using the semi−micro Kjeldahl method. Dissolved organic carbon was determined using deionization leaching (1:5 soil-to-water ratio, *w*/*v*) with an elemental analyzer (FlashSmart, ThermoFisher Scientific, Milan, Italy). N−NH$_4^+$ and N−NO$_3$− were determined using leaching with a KCl solution. Amorphous iron was determined using oxalic acid–ammonium oxalate leaching. Particle composition was determined using the pipette method. Microbial biomass carbon (MBC) and nitrogen (MBN) were determined using chloroform fumigation—potassium sulfate leaching with an elemental analyzer (FlashSmart, ThermoFisher Scientific, Milan, Italy) with leaching coefficients of 0.45 and 0.54 for MBC and MBN, respectively.

The SON components were determined using the semi-micro Kjeldahl nitrogen method. Soil samples were hydrolyzed with 6 M (mol L$^{-1}$) HCl for 12 h to measure SON components [11]. Total acid hydrolyzable N (TAHN), AN, AAN, and (AN+ASN) were determined using 10 M (mol L$^{-1}$) NaOH, 3.5% (*w*/*v*) MgO, a phosphate borate buffer with a pH of 11.2, and a ninhydrin powder oxidation–phosphate borate buffer, respectively. ASN is the difference between ASN+AN and AN. HUN is the difference between TAHN and AN+AAN+ASN. NHN is obtained by subtracting TAHN from STN.

Soil NRA was determined using the phenol-disulfonic acid colorimetric method [29]. Soil DEA was determined via acetylene inhibition [30] using gas chromatography (Agilent 7890B, Agilent Technologies Inc., Palo Alto, CA, USA). Soil PT was determined using the ninhydrin colorimetric method [29]. Soil NAG was determined using the method of Wu et al. [4]: 1.00 g of fresh soil sample was added to 4 mL of 0.1 M (mol L$^{-1}$) acetate buffer (pH 5.5) and 1 mL of 4-MUB-N-acetyl-β-D-glucosaminide solution, set at 37 °C for 1 h.

Then, 1 mL of 0.5 M (mol L$^{-1}$) CaCl$_2$ and 4 mL of 0.5 M (mol L$^{-1}$) NaOH were added. Upon the completion of the reaction, soil NAG was measured with a multifunctional microplate reader (Spectra Max M5, Molecular Devices, Sunnyvale, CA, USA).

### 2.4. Data Analysis

SON components stocks in soil layers of four vegetation types were calculated as follows:

$$\text{Stocks of } SON_i \text{ (Mg ha}^{-1}) = (SON_i \times \text{BD} \times \text{H})/10$$

where $SON_i$ is the content of different SON components (g kg$^{-1}$), BD is the soil bulk density (g cm$^{-3}$), and H is the soil depth (cm).

All variables of vegetation restoration types and soil depths were tested using one-way and two-way ANOVA through a least significant difference test ($p < 0.05$). Values were given as the mean ± the standard deviation (SD). Principal coordinate analysis (PCoA) was used to determine the contents and stocks of the SON component among different vegetation restoration types using R 4.2.1 software (*vegan* package). Linear regression was conducted to analyze the relationship between soil total N and SON components. Redundancy analysis (RDA) was used to determine the relationships between SON components and environmental factors. Statistical analyses were performed using SPSS v24.0. RDA was performed using Canoco5.0 software. Figures were plotted using Origin2020b.

## 3. Results

### 3.1. Soil Mineral N, MBN, and STN Contents

The overall N−NH$_4^+$, N−NO$_3^-$, MBN, and STN were significantly influenced by the vegetation restoration (Figure 2a–d). The N-NH$_4^+$ in AF and SL soils were significantly higher than in GL and CL soils at the 0–10 cm soil depth, while the highest N−NH$_4^+$ at the 10–20 cm soil depth was observed in CL (Figure 2a). The CL soil had the highest N−NO$_3^-$ and STN contents at the two soil depths, which were significantly higher than the other vegetation restoration types (Figure 2b,d). Compared with SL, GL, and CL, the MBN was higher in AF soil by 2.31%–283.46% across the soil depths (Figure 2c).

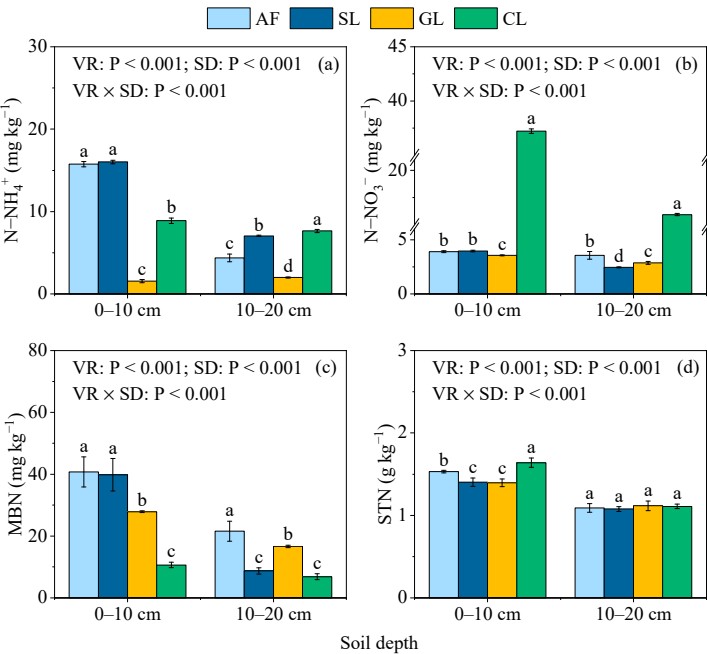

**Figure 2.** Soil N−NH$_4^+$ (**a**), N−-NO$_3^-$ (**b**), MBN (**c**), and STN (**d**) in the selected vegetation restoration types: AF, arbor forest; SL, shrubland; GL, grassland; CL, cropland. VR, vegetation restoration; SD, soil depth; STN, soil total nitrogen. Different lowercase letters indicate significant differences ($p < 0.05$) between vegetation restoration types at the same soil depth.

### 3.2. Contents and Proportion of SON Components

The results of the two-way ANOVA showed that the vegetation restoration type, soil depth, and interaction between the two had significant effects on the SON component contents ($p < 0.05$, Figure 3a–f), except for the effect of soil depth on NHN and the effect of their interaction on HUN. Compared with AF, SL, and CL, the TAHN, AN, and AAN were higher in the GL soil by 1.01%–26.25%, 12.61%–65.67%, and 51.74%–102.61% across the soil depths, respectively (Figure 3a–c). The ASN in the AF and GL soils was significantly higher than in the SL and CL soils at the 0–10 cm soil depth, while the highest ASN at the 10–20 cm soil depth was observed in SL (Figure 3d). The HUN in the AF soil significantly increased by 27.94% and 40.39% at the 10–20 cm soil depth compared with GL and CL, respectively (Figure 3e). In contrast, no significant differences were detected between the vegetation restoration types at the 0–10 cm soil depth. The NHN in the CL soil was significantly higher than the other vegetation restoration types at the two soil depths (Figure 3f).

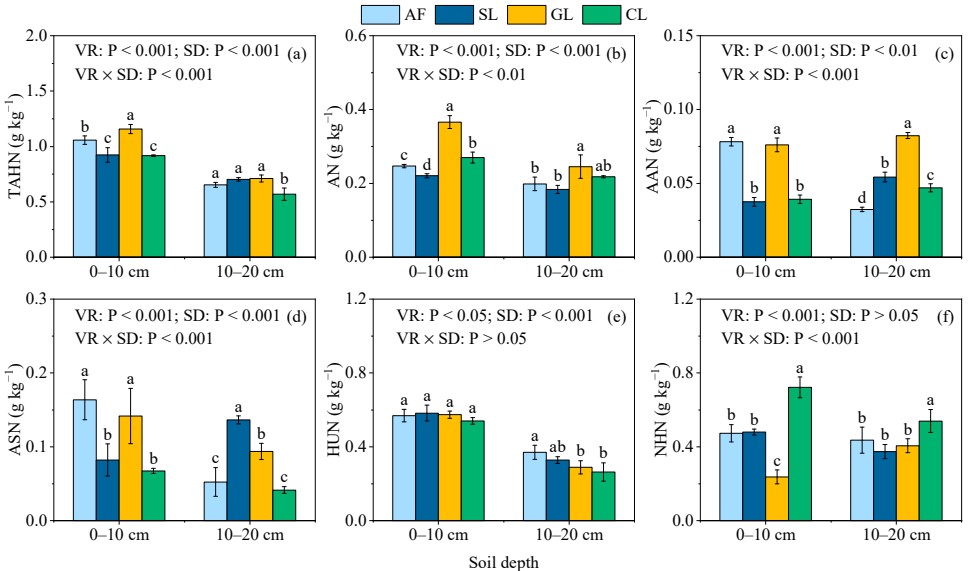

**Figure 3.** SON components in selected vegetation restoration types: (**a**) THAN; (**b**) AN; (**c**) AAN; (**d**) ASN; (**e**) HUN; (**f**) NHN. AF, arbor forest; SL, shrubland; GL, grassland; CL, cropland. VR, vegetation restoration; SD, soil depth. TAHN, total acid hydrolyzable N; ASN, acid sugar N; AN, ammonium N; AAN, amino acid N; HUN, hydrolyzable unidentified N; NHN, non−hydrolyzable N. Different lowercase letters indicate significant differences ($p < 0.05$) between vegetation restoration types at the same soil depths.

The AF, SL, and GL soils had the largest proportion of HUN to STN, while the CL soil had the largest proportion of NHN to STN (Figure 4). AN accounted for 16.13%–26.23% of the STN, with the highest content in GL soil, which was significantly higher than other vegetation restoration types. The soil AAN and ASN accounted for less than 20% of the STN, and the maximum amounts were found in GL and SL soils, respectively. Vegetation restoration significantly altered the proportion and distribution of SON, with an increased proportion of HUN and decreased proportion of NHN.

### 3.3. Stocks of SON Components

A two-way ANOVA indicated that the SON component stocks were influenced by the vegetation type, soil depth, and their interaction ($p < 0.05$, Figure 5a–f), except for the effect of soil depth on the NHNs and the effect of their interaction on the HUNs. The amounts of the TAHNs, ANs, and AANs in the GL soil were significantly greater than the other vegetation restoration types at the two soil depths, except that no significant differences in the TAHNs and AANs were detected between AF and GL (Figure 5a–c). The ASNs in the AF and GL soils were significantly greater than in SL and CL at the 0–10 cm soil depth,

and SL had the greatest ASNs at the 10–20 cm soil depth (Figure 5d). Compared with CL, the HUNs were higher in AF, SL, and GL by 16.87%–47.73% across the two soil depths (Figure 5e). Unlike the TAHNs, ANs, AANs, ASNs, and HUNs, CL had the greatest NHNs at the two soil depths (Figure 5f).

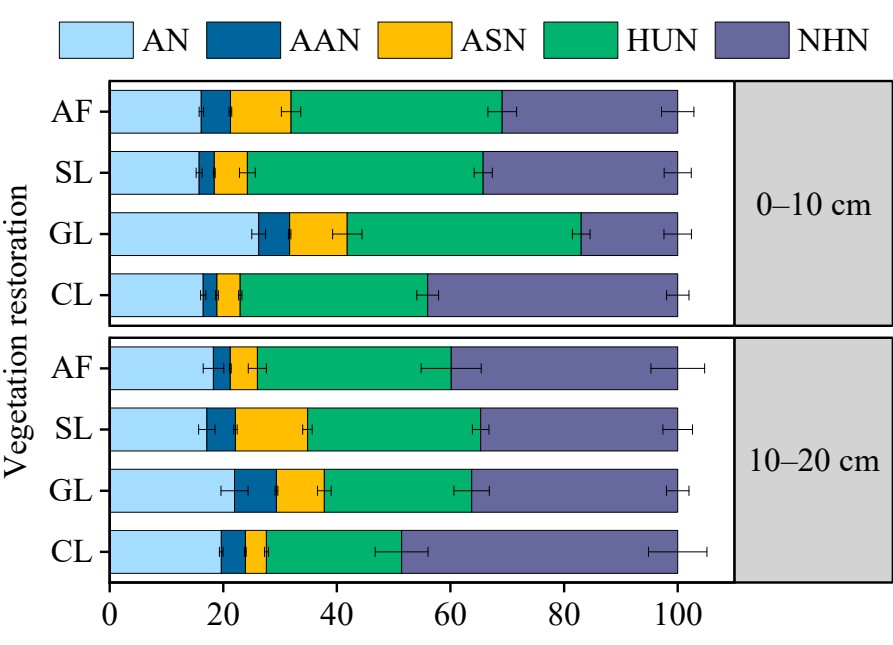

**Figure 4.** Proportion of SON components in selected vegetation restoration types: AF, arbor forest; SL, shrubland; GL, grassland; CL, cropland. ASN, acid sugar N; AN, ammonium N; AAN, amino acid N; HUN, hydrolyzable unidentified N; NHN, non−hydrolyzable N.

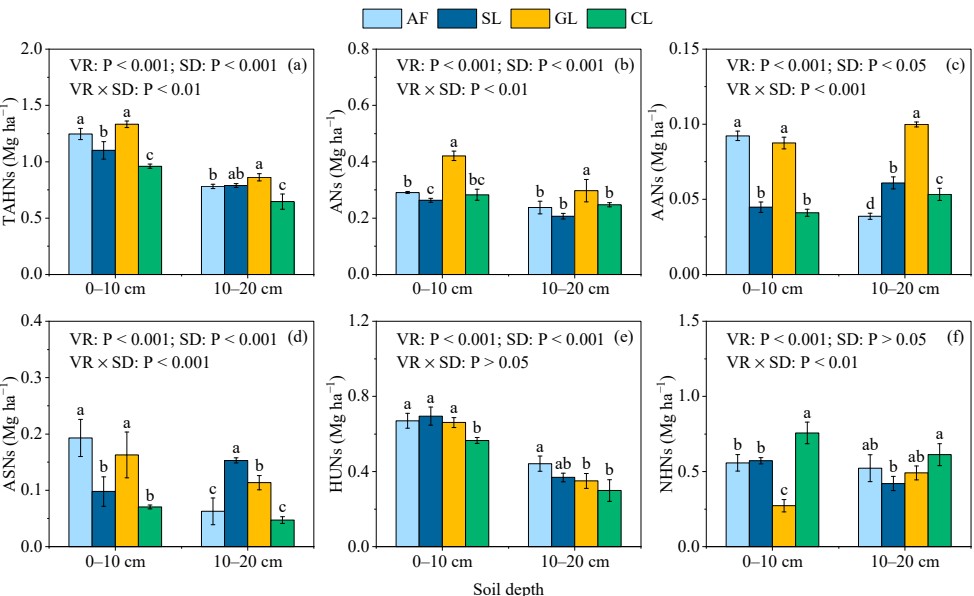

**Figure 5.** Stocks of the SON components in selected vegetation restoration types: (**a**) THANs; (**b**) ANs; (**c**) AANs; (**d**) ASNs; (**e**) HUNs; (**f**) NHNs. AF, arbor forest; SL, shrubland; GL, grassland; CL, cropland. VR, vegetation restoration; SD, soil depth. TAHNs, TAHN stocks; ANs, AN stocks; AANs, AAN stocks; HUNs, HUN stocks; NHNs, NHN stocks. Different lowercase letters indicate significant differences ($p < 0.05$) between vegetation restoration types at the same soil depths.

### 3.4. Soil-N-Cycling Enzyme Activities

Soil−N−cycling enzyme activities in the soil profile varied significantly among the different vegetation restoration types (Figure 6a–d). The NRA activity decreased significantly in the following order: AF > GL > SL > CL (Figure 6a). The DEA and NAG activities in the CL soil were significantly lower than the other vegetation restoration types at the two soil depths; however, no significant differences in NRA between AF, SL, and GL at the 0–10 cm soil depth were found (Figure 6b,d). The AF soil had the greatest PT activity at the 0–10 cm soil depth, followed by SL, GL, and CL (Figure 6c). However, the PT activity in the SL soil was significantly higher than in AF, GL, and CL for the 10–20 cm soil depth.

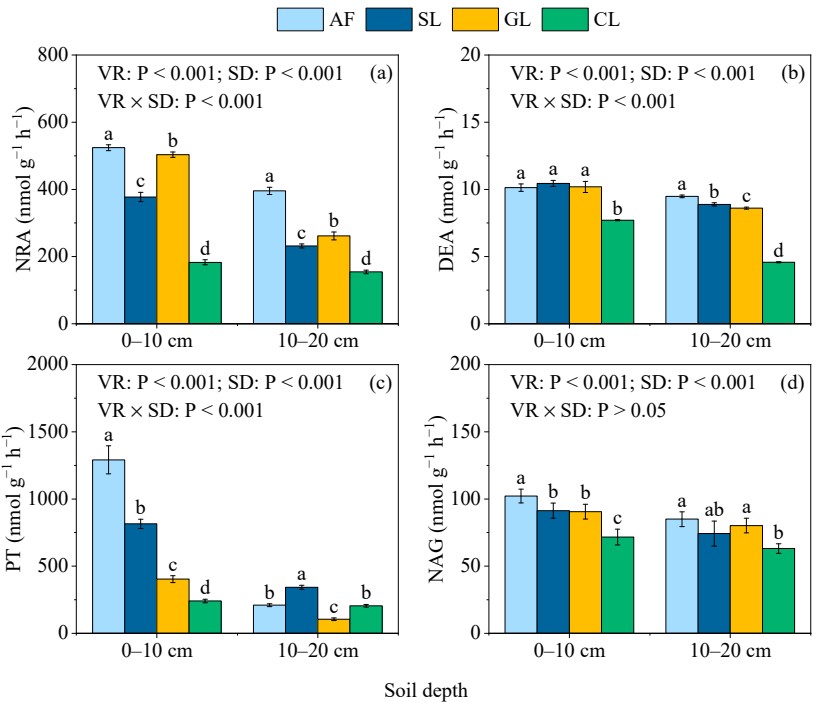

**Figure 6.** Soil−N−cycling enzyme activities in selected vegetation restoration types: (**a**) NRA; (**b**) DEA; (**c**) PT; (**d**) NAG. AF, arbor forest; SL, shrubland; GL, grassland; CL, cropland. VR, vegetation restoration; SD, soil depth. NRA, nitrate reductase; DEA, denitrifying enzyme activity; PT, protease; NAG, β−1,4−N−acetylglucosaminidase. Different lowercase letters indicate significant differences ($p < 0.05$) between vegetation restoration types at the same soil depths.

### 3.5. Factors That Affected the Transformation of Soil N Components

The result of the principal coordinate analysis (PCoA) indicated that the SON component was strongly influenced by vegetation restoration (Figure 7). Redundancy analysis showed that environmental factors explained 98.4% of the variation in the soil N component, with the first two axes explaining 63.71% and 33.05% of the variation, respectively (Figure 8a). $Fe_o$ (54.4%), PT (23.3%), and SOC (9.5%) provided significant and effective contributions to the N component transformation (Figure 8b). $Fe_o$ showed significantly positive correlations with the STN, $N−NH_4^+$, $N−NO_3^-$, and NHN. PT had significantly positive correlations with the STN, $N−NH_4^+$, TAHN, ASN, and HUN. SOC was significantly and positively correlated with the $N−NH_4^+$ and HUN and negatively correlated with the AAN (Figure A1). Overall, the soil physicochemical properties explained more variation in the soil N components than the soil−N−cycling enzyme activity and microbial biomass (Figure 8c).

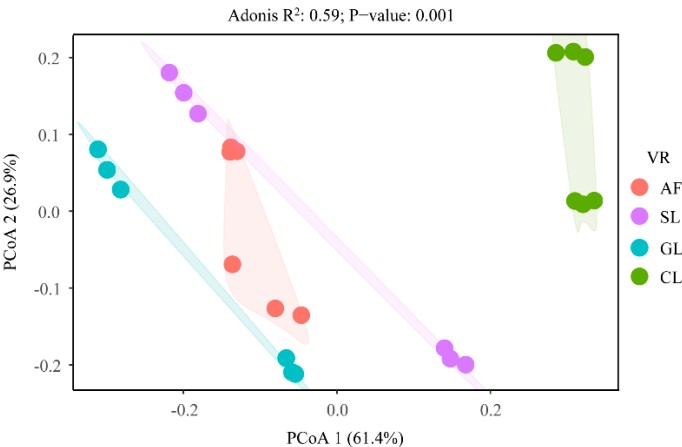

**Figure 7.** Principal coordinate analysis (PCoA) of the SON components in selected vegetation restoration types: AF, arbor forest; SL, shrubland; GL, grassland; CL, cropland. VR, vegetation restoration.

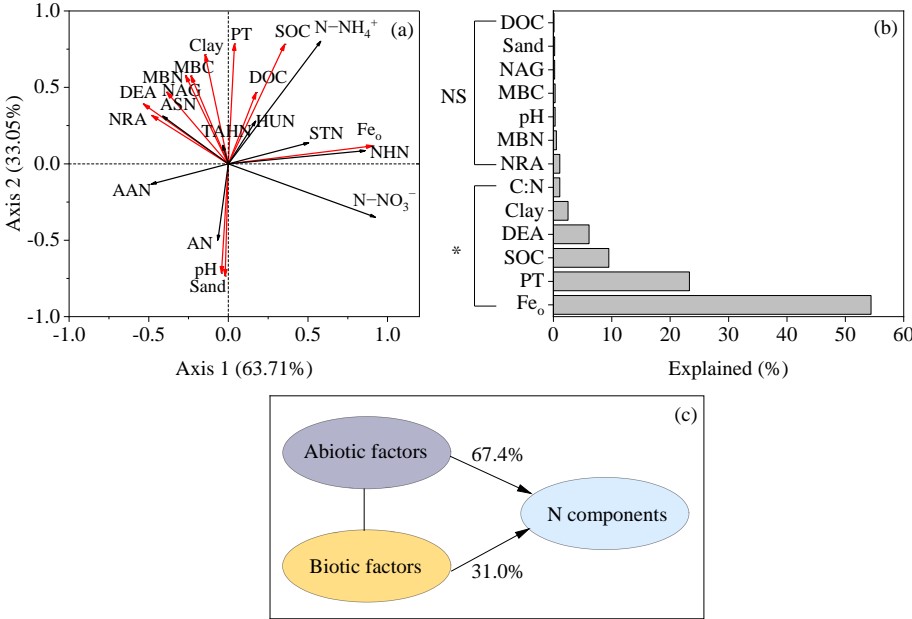

**Figure 8.** Redundancy analysis of the soil N contribution variance (**a**); factor contributions to the soil N components transformation (**b**); and contribution of abiotic and biotic factors for variations in soil N components (**c**): DOC, dissolved organic carbon; SOC, soil organic carbon; C:N, ratio of SOC to STN; $Fe_o$, amorphous iron; MBC, microbial biomass carbon; MBN, microbial biomass nitrogen; NRA, nitrate reductase; DEA, denitrifying enzyme activity; PT, protease; TAHN, total acid hydrolyzable N; ASN, acid sugar N; AN, ammonium N; AAN, amino acid N; HUN, hydrolyzable unidentified N; NHN, non−hydrolyzable N; STN, soil total nitrogen. *, $p < 0.05$; NS, $p > 0.05$.

## 4. Discussion

### 4.1. Effect of Vegetation Restoration on SON Components

SON is both a source and sink of mineral N, which has a direct influence on the amount and rate of soil N mineralization due to differences in its chemical speciation and content. The amount of easily mineralized SON plays a critical role in the N supply capacity of the soil [13]. Generally, vegetation restoration significantly increases the SON component contents and stocks, mainly through plant litter and root secretions that provide N to the soil, increasing the soil N input [26,31]. Additionally, the input of vegetation litter leads to a higher soil C:N (Table 1). This means that microorganisms need to take up more mineral N from the soil to meet their own needs [32], which subsequently increases their ability to use mineral N [33] and results in more microbial assimilation of mineral

N into the SON pool [16,34]. This accelerates the conversion of soil inorganic N to SON, which results in enhancing the soil N storage capacity (Figure 5) and reducing N losses. In agricultural ecosystems, plant litter does not accumulate, which results in less SON source input and the lowest TAHN. Additionally, artificial management practices can reduce the stability of N component cycling in agroecosystems [35]. AN and AAN are the "temporary and slow−acting reservoirs" of SON, which are the main sources of easily mineralized N in soil [12,13]. The GL soil in this study exhibited higher contents and stocks of AN and AAN than the soils with other types of vegetation restoration (Figures 3 and 5). This could be attributed to the shorter life cycle and turnover of herbaceous vegetation, and the ready decomposition and use of herbaceous vegetation litter by soil microorganisms, which rendered a relatively fast rate of soil N cycling [36]. The GL soil was slightly acidic, which led to the proton replacement of some specifically adsorbed cations or the neutralization of negative charges, resulting in a decrease in the adsorption potential of N−$NH_4^+$ [37]. The main components of ASN are synthesized with microbial growth and then accumulated after cell death [12]. In this process, N can be directly fixed and transformed into microbial residues, while being easily decomposed or mineralized into AAN during transformation. This process is also closely related to the turnover of root systems, root secretions, and microbial metabolites [12]. It was shown that ASN is a source of useable N for soil microorganisms when soil N availability is low, but AAN is preferentially utilized over ASN [38]. The increase in ASN was mainly related to MBC. Plant litter increased soil MBC (Table 1), resulting in the accumulation of ASN in the soil [4]. Research showed that vegetation litter is an important resource of soil humus, and N in the litter is mainly in the form of HUN [38]. HUN accounted for 25.92%–41.19% of the STN in this study and its allocated proportion of STN increase was 53.02% (Figure A2), indicating that the HUN was a major contributor to the elevated soil N levels [12]. NHN is refractory to acid hydrolysis to release available N and is often considered to be the unavailable N component. With weathering and microbial processes, these complex N components can gradually decompose to produce soluble SON and provide a long-term N supply [13]. The high proportion of NHN in the present study (16.99%–48.61% of STN) may have been related to the strong leaching and weathering characteristics of soils in the subtropical region, where enriched Fe and Al oxides combine with organic functional groups to form stable N components [39].

### 4.2. Effect of Vegetation Restoration on Soil Cycling Enzyme Activities

NRA, DEA, PT, and NAG are key enzymes involved in soil N transformation, and their activities are closely related to the intensity of soil N transformation and soil N supply capacity [4,14]. In this study, we found that vegetation restoration significantly increased the soil NRA, DEA, PT, and NAG activities. This is consistent with previous studies that confirmed that soil enzyme activities were influenced by vegetation restoration [19,20]. Guan et al. [40] found that the nutrient released by plant residual decomposition improved soil enzyme activities. Vegetation restoration could enhance soil aggregate stability by reducing soil disturbances [7]. Improving the soil aggregate stability could cause the soil environment to be more favorable for soil microbes [16]. We found that vegetation restoration significantly increased the soil microbial biomass, and the trend was consistent with the soil−N−cycling enzyme activities (Figure A3). Vegetation restoration had a varying effect on different soil−N−cycling enzymes (Figure 6). This was mainly because differences in the nutrients released by the plant litter of various vegetation restoration types affected the N cycle products and processes [22,41]. Studies showed that NRA and DEA activities are related to factors such as soil moisture and N source content and availability [42,43]. Compared with forest and grassland ecosystems, soil enzyme activity is lower in agricultural ecosystems because anthropogenic disturbances reduce soil moisture, which then influences the soil microbial metabolic activity [43]. Han et al. [44] found that denitrification of N depended on N−$NO_3^-$ and was strongly correlated with the N-$NO_3^-$ content. N-$NO_3^-$ accumulation in CL soils under the influence of exogenous N

fertilizers (nitrate N fertilizers) over a long period was originally expected to boost NRA and DEA activity. However, low enzyme activities were observed instead. This might have been because $N-NO_3^-$ accumulation led to a significant reduction in the soil pH, which inhibited denitrification in acidic soils [45]. Moreover, the high sand content of agricultural soils and high porosity were not conducive to the growth and reproduction of soil-denitrifying bacteria [42,46]. The increased exogenous N input through the leaching of plant litter during vegetation restoration can promote plant root growth and metabolism, increase root secretion, and accelerate soil microbial colonization, thus activating soil N mineralization and decomposition enzyme activities (PT and NAG) [4,40]. Additionally, higher PT and NAG activities promote the decomposition of plant litter, forming a positive feedback mechanism.

*4.3. Factors That Influenced the Soil N Transformation*

The results of the redundancy analysis showed that the main factors that influenced the soil N transformation in the DDR were $Fe_o$, PT, and SOC. The study area was in the southern subtropical monsoon climate zone, where the soil suffers severe leaching and weathering, experiences desiliconization, and is rich in Fe. Therefore, iron oxides play important roles in the soil ecosystem [47]. Iron oxides may influence the transformation of SON components in the soil ecosystem in two ways: (1) As an inorganic cementing substance, iron oxides can bind to minerals to form nanoparticles, which further cement to form microaggregates and macroaggregates [48]. The aggregates have a large specific surface area and soil N strongly adsorbs to them. Therefore, they can effectively block the capture of organic molecules by extracellular enzymes and heterotrophic microorganisms [49] and reduce soil N loss [8]. In this study, $Fe_o$ showed a significant positive correlation with STN, indicating that $Fe_o$ played an important role in maintaining high levels of soil N. (2) The iron redox system is important in subtropical soils, and iron oxides are both electron acceptors and donors that participate in the biochemical processes of soil N [47]. For example, FeO participates in the Fenton reaction to produce reactive oxygen species that can cleave recalcitrant organic molecules and increase the biological availability of soil organic matter [49]. $Fe_o$ has a greater surface activity energy than other iron oxides and has a greater ability to adsorb and retain soil nutrients [49,50]. However, due to its poor crystallinity, an external environmental disturbance can disrupt the unstable mineral lattice, allowing the adsorbed N to be released and increasing soil N availability. Therefore, $Fe_o$ acts as temporary storage for soil N, reducing the loss of N due to soil erosion and releasing adsorbed N for plant uptake when necessary.

Previous studies generally concluded that the rate-limiting step in SON conversion is the decomposition of proteins into oligopeptides and amino acids by extracellular proteases, suggesting that PT plays an important role in the soil N cycle [4,17,51]. PT is involved in the turnover of the active SON pool, such as ASN and AAN [13,38]. Additionally, the PT activity also showed a positive correlation with the HUN content. HUN accounted for a relatively large proportion of TAHN and the highest allocation to STN, suggesting that PT activity may determine soil N availability levels through the mineralized decomposition of HUN. Gerard [52] used a meta-analysis to determine that soil N component conversion was closely related to soil organic carbon. Farrell et al. [53] showed that the conversion and uptake of SON by soil microbes were influenced by carbon availability and that the potential mineralization capacity of SON was also limited by soil carbon resources [54]. Increased exogenous organic carbon from vegetation restoration provides sufficient energy for N transformation, which directly affects the soil microbial biomass and activity [8,55], thereby regulating the content and availability of all forms of soil N. This reveals that plant–organic carbon–soil microbial coupling drives the "carbon–nitrogen" cycle in ecosystems.

## 5. Conclusions

Our results revealed that vegetation restoration significantly affected soil N availability by altering organic nitrogen components. Vegetation restoration increased the active soil

organic nitrogen components (AN, AAN, and ASN) and decreased the proportion of non−hydrolyzable nitrogen components. The practice of returning cropland to grassland was more favorable to the increase in organic nitrogen levels and availability overall. In addition, vegetation restoration increased the nitrate reductase, denitrifying enzyme, proteases, and β−1,4−N−acetylglucosaminidase activities and increased the soil microbial biomass carbon and nitrogen contents. Soil N transformation was achieved under the joint influence of chemical and biological processes, and the amorphous iron, protease activity, and organic carbon contents were the key influencing factors of N transformation. This indicated that vegetation restoration in the Danxia degraded region could "mineralize and fixate" organic matter through amorphous iron and protease to change the organic N components, and thus, affect the soil N supply and availability.

**Author Contributions:** Conceptualization, C.W., Q.Y. and K.L.; methodology, C.W., X.Z. and J.C.; writing—original draft preparation, C.W.; writing—review and editing, C.Z., B.Z. and X.L. All authors have read and agreed to the published version of the manuscript.

**Funding:** This work was supported by the Natural Science Foundation of Guangdong (2021A1515011543), the Guangdong Education Science 13th Five-Year Plan Project (2020GXJK116), and the Key Project of Natural Science of Guangzhou Xinhua University (2020KYZD02).

**Data Availability Statement:** Not applicable.

**Acknowledgments:** We thank Qiong Ye for providing the location map of Danxia Mountain, and reviewers and the editors for their reviews of this manuscript.

**Conflicts of Interest:** The authors declare that the research was conducted in the absence of any commercial or financial relationships that could be construed as a potential conflict of interest.

## Appendix A

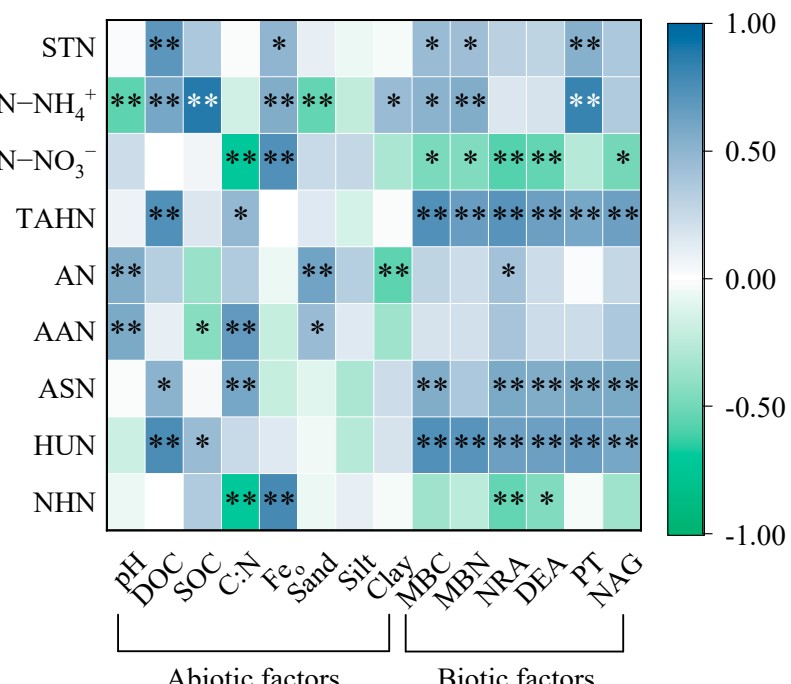

**Figure A1.** Correlation between the abiotic–biotic factors and soil N components: TAHN, total acid hydrolyzable N; ASN, acid sugar N; AN, ammonium N; AAN, amino acid N; HUN, hydrolyzable unidentified N; NHN, non-hydrolyzable N; STN, soil total N; DOC, dissolved organic carbon; SOC, soil organic carbon; C:N, ratio of SOC to STN; $Fe_o$, amorphous Fe; MBC, microbial biomass carbon; MBN, microbial biomass nitrogen; NRA, nitrate reductase; DEA, denitrifying enzyme activity; PT, protease; NAG, β-1,4-N-acetylglucosaminidase. *, $p < 0.05$; **, $p < 0.01$.

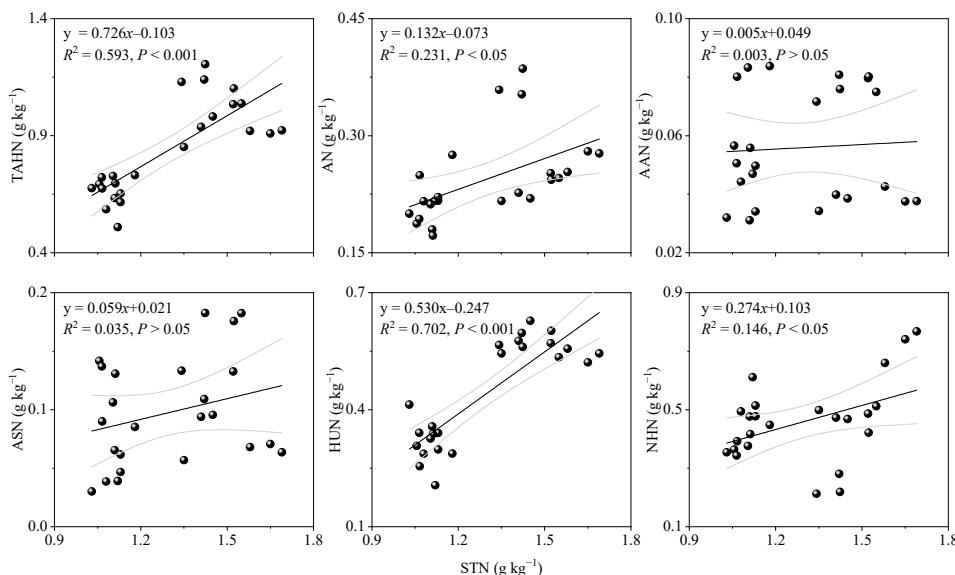

**Figure A2.** Relationship between the STN and SON components: TAHN, total acid hydrolyzable N; ASN, acid sugar N; AN, ammonium N; AAN, amino acid N; HUN, hydrolyzable unidentified N; NHN, non-hydrolyzable N; STN, soil total nitrogen.

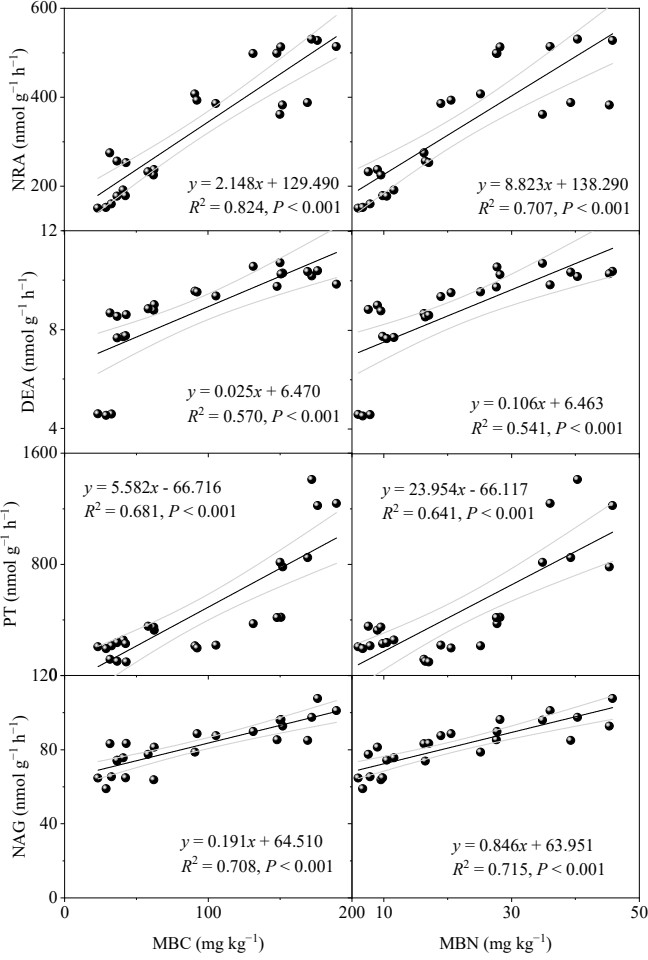

**Figure A3.** Linear analysis between the soil microbial biomass (MBC and MBN) and extracellular enzyme activities (NRA, DEA, PT, and NAG): MBC, microbial biomass carbon; MBN, microbial biomass nitrogen; NRA, nitrate reductase activity; DEA, denitrifying enzyme activity; PT, protease; NAG, $\beta-1,4-N-$acetylglucosaminidase.

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
