# Peer review of "Soil Organic Nitrogen Components and N−Cycling Enzyme Activities Following Vegetation Restoration of Cropland in Danxia Degraded Region"

_forests, doi:10.3390/f13111917_

Round 1

Reviewer 1 Report

it was an honor to have the opportunity to review the manuscript. This study focused on Soil organic nitrogen components and N cycling enzyme activities in Danxia degraded region. It is an interesting study. However, there are some problems needed to be solved before the consideration for acceptance.

(1)    Compared with a scientific research paper, this article is more like an experimental report. From the author's analysis and writing, I can't see the significance and necessity of the research. The authors need to conduct in-depth analysis of their experimental data to reveal the purpose and significance of this research, rather than just conducting data.

(2)    In the introduction, there is no need to explain the various components of hydrolysis N. More emphasis should be placed on the research status, frontiers and problems of related research fields, and leads to scientific questions.

(3)    Detailed introduction to sampling methods

(4)    Table 1, use analysis of variance to mark the differences in soil indicators between different vegetation types.

(5)    There are only four planting quilt types. It is not recommended to emphasize that there are significant differences between different vegetation types, you can You can directly have different vegetation types.

Reviewer 2 Report

This is quite inetresting manuscript which reflects soil biological dynamica in forests and grasslands of China. Nevertheless minor pedological suggestions are mande:

1. Granulometric composition shoul be in percents, but not in g/kg, this is serious mistake from pedological point of view.

2.  I reccomend to provide insert map of the study plots location. China is to big country and broad audience of readers shoud understand where the study was conducted on the map.

3. I reccomend to provide soil Taxonomy, sequence of soil horizons in typical soil sections and soil sections pictures.

4. I reccomend to  provide names of landscapes.

5. I reccomend to provide climatic and hydrological conditions of the studied plots, cause they a key drivers of biological activity.

6. The definition of biotic and abiotic factors in this paper is not clear, I reccomend to clarify it in terms of ecology. 

Overall reccomendation - to make paper more pedological and ecological in terms of interpretation of results. Now, there is skewness to laboratory desing and results. While the journal is Forests more idea about relation of your data to forest biogenic processes should be provided. 

Reviewer 3 Report

This paper analyzes the effect of returning farmland to forest on the geochemical cycle of soil organic nitrogen. The results proved that vegetation restoration in the DDR changed the SON components through the “mineralization-fixation” of organic matter by amorphous iron and proteases, which in turn affected the level of available soil N. This is a good story, and organized well, I have no further comments, just some small suggestions.

Specific suggestions show below:

Keywords: change ‘N cycling enzymes’ to ‘enzymes’, separate ‘Danxia degraded region’ to ‘Danxia’ and ‘soil degradation’, and delete ‘N availability’

Introduction: This part is very detailed, and the details are a bit like textbook knowledge. Lack of new research status and progress. It is suggested that the conceptual description should be omitted and a literature review should be added. Especially the second and fourth paragraphs.

Materials and Methods:

Table 1: what is AG in Table 1? Is it GL?

Table 1: what are the values mean? Mean ± SD? Mean ± SE? Please make it clear.

Table 1: please add ANOVA results to give readers more information.

Line 149. Water-soil ratio was 2.5:1, volume ratio or weight ratio? Same as Line 153. You can refer to the Line 164 in your MS.

Line 162. M is the unit of weight, not a concentration unit. So, you need to note it. I totally know this is a verbal error that is often made in traditional Chinese teaching, but it cannot be made in scientific expression. Mole is the unit of quantity of a substance, and mole per liter is the unit of concentration. Not to be confused. Change it as 6 M (mol L-1) HCl.

Line 173. CaCl2, 2 needs to be subscripted.

Line 180. What does Ma mean in the formular?

Results:

In your figures, what are VR and SD mean? Note these in your figure caption.

I find the abbreviations in your full text a bit confusing, AF and AG for a while, SF and GL for a while. Too careless!

I am interested in Figure 2 and Figure 4. What is the difference between 3.2 and 3.3? How did you think about it? is it necessary? I hope you can answer it in the response letter.

Discussion and Conclusion: No further comments, well done.

Reviewer 4 Report

Through the analysis of SON and enzyme activity, this paper plays a guiding role in the nitrogen cycle process in the DDR area. The description and method described in this paper are feasible, but they have not yet reached the publication standard, and it is recommended to accept them after modification.

1. The introduction is too complicated and should focus on the main scientific issues. The conceptual description can be expressed in the way of references.

2. In the introduction and description process, it is suggested to increase the research progress in non-DDR areas, focusing on the action mode of enzymes and the products produced during the nitrogen cycle.

3. Line 104-110, it is suggested to simplify the scientific problems to two.

4. Line 143 Supplement the differences between indicators

5.line 131 Please state the number of samples collected

6.line275 Please explain how the contribution rate described in the text is calculated

7 It is suggested that the first two parts of the discussion be merged while adding a discussion of the effects of vegetation on enzyme cycling.

8. It is suggested to delete the conclusion section "These results provide a scientific basis for improving the characteristics of soil N components, promoting soil intrinsic N cycling, and restoring soil quality in the Danxia degraded region."

9. Though English language seems is in general understandable (I’m not a native English), I recommend a carefully review to avoid errors and inconsistencies.

Round 2

Reviewer 1 Report

The authors have revised the article and suggest acceptance.

Reviewer 4 Report

I believe that authors has incorporated all the comments and response to reviewers comments are satisfactory.